# The Characteristics of 206 Long-Term Survivors with Peritoneal Metastases from Colorectal Cancer Treated with Curative Intent Surgery: A Multi-Center Cohort from PSOGI

**DOI:** 10.3390/cancers13122964

**Published:** 2021-06-13

**Authors:** Yasuyuki Kamada, Koya Hida, Yutaka Yonemura, Paul H. Sugarbaker, Shadin Ghabra, Soichiro Ishihara, Hiroshi Nagata, Koji Murono, Takanori Goi, Kanji Katayama, Mitsuhiro Morikawa, Beate Rau, Pompiliu Piso, Miklos Acs, Federico Coccolini, Emel Canbay, Mao-Chih Hsieh, Aditi Bhatt, Pierre-Emmanuel Bonnot, Olivier Glehen

**Affiliations:** 1Department of Surgery, Graduate School of Medicine, Kyoto University, Kyoto 6068507, Japan; hidakoya@kuhp.kyoto-u.ac.jp; 2Department of Regional Cancer Therapy, Peritoneal Surface Malignancy Center, Kishiwada Tokushukai Hospital, Kishiwada 5960042, Japan; y.yonemura@coda.ocn.ne.jp; 3Department of Regional Cancer Therapy, Peritoneal Surface Malignancy Center, Kusatsu General Hospital, Kusatsu 5258585, Japan; 4Center for Gastrointestinal Malignancies, MedStar Washington Hospital Center, Washington, DC 20010, USA; Paul.Sugarbaker@medstar.net (P.H.S.); shadin.ghabra@medstar.net (S.G.); 5Department of Surgical Oncology, The University of Tokyo, Tokyo 1138654, Japan; soichiro.ishihara@gmail.com (S.I.); hiroshin84@gmail.com (H.N.); MURONOK-SUR@h.u-tokyo.ac.jp (K.M.); 6First Department of Surgery, Faculty of Medicine, University of Fukui, Eiheiji 9101193, Japan; tgoi@u-fukui.ac.jp (T.G.); kanji@g.u-fukui.ac.jp (K.K.); mmitsu@u-fukui.ac.jp (M.M.); 7Department of Surgery, Campus Virchow-Klinikum and Charité Campus Mitte, Charité—Universitätsmedizin Berlin, 13353 Berlin, Germany; beate.rau@charite.de; 8Department for General and Visceral Surgery, Barmherzige Brüder Hospital, 93049 Regensburg, Germany; Pompiliu.Piso@barmherzige-regensburg.de (P.P.); Miklos.Acs@barmherzige-regensburg.de (M.A.); 9Department of General, Emergency and Trauma Surgery, Pisa University Hospital, 56100 Pisa, Italy; federico.coccolini@gmail.com; 10Department of General Surgery, İstanbul University İstanbul School of Medicine, İstanbul 34093, Turkey; drecanbay@gmail.com; 11Department of General Surgery, Wan Fang Hospital, Taipei Medical University, Taipei 116, Taiwan; 86091@w.tmu.edu.tw; 12Department of Surgical Oncology, Zydus Hospital, Thaltej, Ahmedabad 380054, India; aditimodi31@gmail.com; 13Department of Surgical Oncology, Centre Hospitalier Lyon-Sud, 69310 Lyon, France; pierre-emmanuel.bonnot@chu-lyon.fr (P.-E.B.); olivier.glehen@chu-lyon.fr (O.G.)

**Keywords:** peritoneal metastasis, colorectal cancer, long-term survivors, cytoreductive surgery, HIPEC

## Abstract

**Simple Summary:**

Cytoreductive surgery (CRS) combined with hyperthermic intraperitoneal chemotherapy improves survival in selected patients with peritoneal metastases from colorectal cancer (CRC). However, the characteristics of long-term survivors are not well documented. This study set out to investigate the patient characteristics associated with the long-term survival of peritoneal metastases from CRC. We retrospectively analyzed 206 long-term survivors who underwent CRS for peritoneal metastases from CRC. We found that most long-term survivors showed low peritoneal cancer index (PCI), low PCI of small bowel subsets, and complete cytoreduction (CC-0), while some exhibited characteristics considered associated with poor prognosis.

**Abstract:**

Background: We conducted this study to review the patient characteristics associated with long-term survival in patients with peritoneal metastases from colorectal cancer who underwent cytoreductive surgery (CRS). Methods: We retrospectively investigated patients with peritoneal metastases from CRC treated with curative intent surgery with or without hyperthermic intraperitoneal chemotherapy at 13 institutions worldwide between January 1985 and April 2015 and survived longer than five years after the first CRS for peritoneal metastases. Clinical and oncological features and therapeutic parameters were described and analyzed. Results: Two hundred six long-term survivors were available for study. The median peritoneal cancer index (PCI) of this cohort was 4 (interquartile range (IQR), 2–7), and the median score of the small bowel regions of the PCI (SB-PCI) was 0 (IQR, 0–2). Complete cytoreduction (CC-0) was achieved in 180 (87.4%) patients. Recurrence was observed in 122 (59.2%) patients at a median of 1.8 (IQR, 1.2–2.6) years. Conclusions: While most long-term survivors showed low PCI/SB-PCI and CCR-0, some had characteristics considered associated with poor prognosis. Curative intent treatments may be considered in well-informed and fit patients showing negative factors affecting survival outcome.

## 1. Introduction

In colorectal cancer (CRC) patients, peritoneal metastases are observed in 5–10% at the time of treatment for primary cancer (i.e., synchronous peritoneal metastases) and in 15–30% of the follow-up after primary cancer surgery (i.e., metachronous peritoneal metastases) [1,2,3]. More than 20 years ago, peritoneal metastases from CRC were considered terminal conditions and treated with palliative treatment. Even with the development of chemotherapy in the last two decades, the treatment efficacy against peritoneal metastases from CRC has been limited [4,5,6,7]. For more than two decades, the combination of cytoreductive surgery (CRS) and hyperthermic intraperitoneal chemotherapy (HIPEC) has improved survival outcomes of CRC peritoneal metastases [7,8,9,10,11,12,13,14]. The aim of CRS is to remove all macroscopic tumor nodules directly, with peritonectomy or with visceral resections. The aim of HIPEC is to eradicate any residual disease with the administration of heated chemotherapy after completing cytoreduction. With these treatments, the number of patients with peritoneal metastasis from CRC who achieve long-term survival has increased over the past decade [14,15]. However, the combination of CRS and HIPEC is a radical treatment associated with established mortality and morbidity [16,17,18]. The patients must be carefully selected to prevent complications and attain survival benefits.

Although the current literature addresses the factors related to survival outcomes in detail [19,20,21,22], little has been reported on factors associated with long-term survival in peritoneal metastases from CRC. Therefore, the purpose of this study is to describe the long-term survivors’ characteristics of a large cohort in patients with peritoneal metastases from CRC who underwent curative intent CRS. We present a collaborative research of the Peritoneal Surface Oncology Group International (PSOGI) to identify the characteristics of long-term survivors.

## 2. Results

### 2.1. Patient Characteristics

In the 11 PSOGI group hospitals and 2 Japanese hospitals, 1455 patients underwent primary CRS with or without HIPEC for CRC peritoneal metastasis (Appendix A). Among the 1455 patients, 206 (14.2%) were identified as long-term survivors and included in this study (Figure 1). In 84 of the 206 patients, a cure was observed. The median follow-up period was 6.6 (range, 5.0–28.6) years. Among the 13 institutions, one institution recorded 83 patients, 2 recorded 23 and 21 patients, and the others recorded fewer than 20 patients (Appendix A).

Table 1 summarized the baseline characteristics of this cohort. One hundred one male patients (49.0%) and one hundred five female (51.0%) were included, with a median age of 58 years (interquartile range (IQR), 49–66). Of the 206 patients, the primary tumor sites were the right colon in 90 patients (43.7%), the left colon in 101 (49.0%), and the rectum in 14 (6.8%). Regarding the onset, 89 patients (43.2%) had synchronous metastases, whereas 93 patients (45.1%) had metachronous metastases. One hundred forty-nine patients (72.3%) were histologically diagnosed as well to moderately differentiated adenocarcinoma, fifty patients (24.3%) as mucinous, and six patients (2.9%) as poorly differentiated and/or signet ring cell. Lymph node metastases were pathologically proven in 123 patients (59.7%). Among the long-term survivors with lymph node metastases, peritoneal metastases occurred synchronously in 62 (50.4%), metachronously in 56 (54.5%), and unknown in 5 (4.0%). In 27 patients (13.1%), liver metastases were detected and resected.

The median peritoneal cancer index (PCI) of the 206 patients was 4 (IQR, 2–7). Categorizing PCI in this cohort, 169 patients (82.0%) had PCI ≤ 10, 23 (11.2%) had PCI 11–20, and 4 (3.1%) had PCI ≥ 21. The median score of small bowel regions of the PCI (SB-PCI) was 0 (IQR, 0–2). One hundred thirty patients (63.6%) presented with SB-PCI = 0, 51 patients (24.8%) with 1–4, and 9 patients (4.4%) with ≥5.

### 2.2. Treatment Factors

The data in Table 2 provides details of the treatment factors in the 206 long-term survivors. One hundred thirty-seven patients (66.5%) received preoperative systemic chemotherapy; 114 patients received oxaliplatin- or irinotecan-based regimens, and 67 received additional anti-VEGF or anti-EGFR treatment. Preoperative chemotherapy of more than six cycles was performed in 29 patients (14.1%).

Complete cytoreduction (CC-0) was achieved in 180 patients (87.4%), CC-1 in 22 patients (10.7%), and CC-2 in 2 patients (1.0%). One hundred fifty-one patients (73.3%) received HIPEC. Among 24 patients with CC-1/2, HIPEC was performed in 21 patients (87.5%), intraperitoneal chemotherapy in 7 patients (29.2%), postoperative systemic chemotherapies in 9 patients (37.5%), and no treatments in 8 patients (33.3%). Technical variety included exposure technique (open versus closed), duration, and temperatures (40.0 to 43 °C). Mitomycin-based regimens were used in 85 and oxaliplatin-based regimens in 63 patients.

Major complications (grade ≥ IIIA) according to the Clavien-Dindo classification occurred in 38 patients (18.4%): intra-abdominal in 31 patients and extra-abdominal in 7 patients. Postoperative adjuvant systemic chemotherapy was performed in 149 patients (72.3%).

### 2.3. Long-Term Outcomes

The site of relapse and treatment for recurrence are presented in Table 3. Tumor recurrence occurred in 122/206 cases (59.2%), and the time to recurrence was a median of 2.0 years (95% confidence interval, 1.7–2.1). The sites of recurrence included isolated peritoneum (*n* = 43), liver (*n* = 12), abdominal wall (*n* = 11), lung (*n* = 9), lymph nodes (*n* = 5), bone (*n* = 1), and multiple sites (*n* = 41). In this group of 122 patients with recurrence, 70 patients were treated with second CRS and/or metastasectomy with or without HIPEC.

## 3. Discussion

This retrospective, international multicenter study shows a cohort of long-term CRC survivors with peritoneal metastases treated with CRS combined with HIPEC. The aim of this study is to present these rare patients and describe their characteristics. Among the initial cohort of 1455 patients who underwent CRS for peritoneal metastases, 206 patients survived beyond five years, and 84 of 206 patients remained recurrence-free more than five years after the first CRS. This study is the largest series in the world, conducted in 13 different institutions from eight countries, which focused on the clinical and oncological features of long-term survivors in CRC peritoneal metastases.

Although many centers worldwide adopt CRS and HIPEC for peritoneal metastases, there is room for debate about this combined treatment. One criticism for these procedures is the uncertainty about the effectiveness of HIPEC for peritoneal metastases from CRC; the PRODIGE-7 trial questioned the role of HIPEC with oxaliplatin in the clinical management of peritoneal metastases from CRC [23]. In our study, 55 patients of the 206 long-term survivors (26.7%) did not receive HIPEC, establishing that HIPEC is not essential for long-term survival. Whether CRS and HIPEC may add to long-term survival as compared to CRS alone cannot be determined from our data. Another criticism is the high morbidity and mortality rate associated with this procedure. It has been reported that the morbidity rates ranged from 23% to 44%, and mortality rates ranged from 0% to 12% [24]. However, a recent study reported that morbidity and mortality rate after CRS/HIPEC is decreasing due to establishing the surgical procedure and patient selection [25]. Postoperative complications may affect long-term survival [26], and our study reported a relatively low rate of major postoperative complications comparing to other reports in literature. Patient selection to identify candidates for the radical procedure is mandatory for improved outcomes.

It has been reported that patients with peritoneal metastasis treated with modern systemic chemotherapy have a median overall survival (OS) of at least 22 months [27]. On the other hand, CRS/HIPEC is considered to improve a survival outcome for patients with peritoneal metastasis from CRC; the median OS is 30–43 months [8,28,29]. However, studies on CRS and HIPEC have several limitations: small sample sizes, heterogeneity of patients, and lack of control groups. Accordingly, previously published guidelines contain weak recommendations based on low-quality evidence [30,31]. In this study, we presented the characteristics of long-term survivors who underwent CRS. Although it does not indicate the survival benefit of CRS and HIPEC as compared to modern systemic chemotherapy, our study showed many long-term survivors and cured patients in CRS performed in highly selected patients.

Additional valuable information was acquired in this study. First, we ascertained the PCI-distribution of long-term survivors in peritoneal metastases from CRC. Most of the patients in our cohort (169/206, 82.0%) exhibited a PCI ≤ 10 for a median of 4 (IQR, 2–7). Our recently published study, which used data on patients with peritoneal metastases from CRC in two Japanese hospitals, compared the characteristics of long-term survivors with those of non-survivors (OS < 5 years) [32]. The previous study showed that the median PCI was significantly lower in long-term survivors (4 (range, 1–27) versus 9 (range, 0–39), *p* < 0.001), and the cohort showed the following results: the 5-year survival rate was 14.0%; the median PCI was 8 (IQR, 3–20); the distribution of the PCI was 0 to 5 in 86 patients, 6 to 10 in 50, 11 to 15 in 27, 16 to 20 in 21, and ≥21 in 52. The PCI provides a quantitative assessment of the peritoneal disease extent and has been reported to be associated with OS [8,33,34,35]. The assumption that there is a strong association between PCI and completeness of CRS is now widely accepted. Several investigators have suggested that CRS and HIPEC should not be offered in patients with peritoneal carcinomatosis from CRC when the predicted PCI is >17–20 [8,35]. In our cohort, 1.9% (4/206) showed a PCI > 20, and additionally, in the subgroup of cured patients, 3.6% (3/84) presented with a PCI > 10, and no patient had a PCI > 20. Also, about 90% of the long-term survivors and cured patients (180/206, 87.4%, and 77/84, 91.7%) achieved CC-0. The previous study [32] demonstrated that there was a statistically significant difference in CC-0 rates between long-term survivors and non-survivors (33/33 (100%) versus 141/203 (69.8%), *p* < 0.001). As such, these data indicate that low PCI and CC-0 are associated with long-term survival and cure in patients with peritoneal metastases from CRC, although a statistical comparison should not be conducted because of the different datasets. On the other hand, some of the long-term survivors presented with high PCI and/or CC-1/2 in this study. If the curative intent treatments are considered reasonable, patients with these adverse prognostic factors would not necessarily have to be excluded.

Second, the SB-PCIs were also low; the median SB-PCI was 0 (IQR, 0–2). It is generally accepted that the small bowel’s involvement is associated with poor prognosis and a cause of incomplete cytoreduction, particularly when the peritoneal tumors are located in the junction between mesentery and the small bowel [36,37,38]. Of note, more than half (63.1%, 130/206) of the patients showed an SB-PCI = 0. In our previous study [32], the median SB-PCI for the entire cohort was 2 (IQR, 0–3). Between this present study and the previous one, the SB-PCI cannot be statistically compared, but these findings suggest that lower SB-PCIs than that of other abdominopelvic regions are needed for long-term survival. Disease extension to the small bowel areas may be a future relative contraindication for this treatment.

Third, some of the patients having factors associated with poor prognoses achieved >5-year survival (e.g., liver metastases [39,40], signet ring cell carcinoma [40,41,42,43], rectal primary [44,45,46], incomplete cytoreduction [8,47,48]). Notably, patients with lymph node metastases at the time of primary tumor resection constituted more than one-half of the long-term survivors and the subgroup of cured patients (123/206, 59.7% and 51/84, 60.7%). In previous studies, lymph node metastases were found to be predictive factors of poor prognosis [8,49,50,51]. The prognostic nomogram (COMPASS) also includes pathological nodal status among the four clinical factors [52]. However, with standardization of total mesocolic excision, we can remove tumors en bloc with lymphatic, decreasing local recurrence. Our study’s results raise an obvious possibility that patients with lymph node metastases could accomplish long-term survival and cure.

Our study’s strengths are the large number of patients described, the number of international institutes participating, and the novelty focusing on long-term survivors’ characteristics. The present study can generate new research questions and form hypotheses for research concerning CRC peritoneal metastases, whose outcome is long-term survival.

This study has several limitations for various reasons. First, long-term survival and cure are not formally defined and based only on survival times from previous studies [14,53]. Second, because of the nature of a multi-institutional retrospective study, we have the observation and treatment variability. In addition, there were several missing data which may cause selection biases. Finally, as a retrospective descriptive research, our present study lacked any comparison of control groups for statistical analysis of the effectiveness of CRS/HIPEC and prognostic factors. We could not collect data on all patients with colorectal peritoneal metastases who underwent CRS/HIPEC in the 13 institutions. Therefore, we did not compare the characteristics between long-term survivors and non-survivors. However, this study’s data permit a detailed assessment of long-term survivors’ clinical features in patients with CRC peritoneal metastases.

## 4. Materials and Methods

This is a multicentric, retrospective study of patients diagnosed with peritoneal metastases from CRC treated with CRS between 1985 and 2015. The study was conducted at 13 different institutions from Japan (*n* = 5), Germany (*n* = 2), France (*n* = 1), the United States (*n* = 1), Italy (*n* = 1), Turkey (*n* = 1), Taiwan (*n* = 1), and India (*n* = 1). The query was performed to each institution in January 2020. We identified 1455 patients who were initially diagnosed with CRC peritoneal metastases and received CRS with or without HIPEC. From these initial 1455 patients diagnosed with CRC peritoneal metastases and received CRS with or without HIPEC, long-term survivors were identified as those who had an OS of ≥5 years after curative intent CRS for peritoneal metastasis from CRC. We defined “cure” as a recurrence-free survival (RFS) more than five years after the CRS’s date and considered cured patients as the subgroup of the study population. Recurrence was confirmed either on pathological or radiologic findings when peritoneal nodules are detected or increased in size. Since there is no official definition of long-term survival after CRS, we used an OS of ≥5 years as our criteria.

Patients who underwent chemotherapy alone and patients with peritoneal metastases from appendiceal carcinoma were excluded.

### 4.1. Study Protocol

We obtained standard data on the patient’s status before the treatment, tumor characteristics, and treatment details. All patients in the long-term survivor cohort had histologically confirmed tumor pathology of peritoneal metastases from CRC.

Primary tumors located in the cecum, ascending colon, and transverse colon were defined as right-sided colon cancer. Those located in the splenic flexure, descending colon, and sigmoid colon were defined as left-sided colon cancer. Rectum was considered as a separate entity from colon. Lymph node involvement at the time of primary tumor resection was defined by a positive histologic diagnosis of lymph node metastases. According to institutional preference, HIPEC was performed either using an open or closed technique, with the target temperature ranging from 40.0 to 43.0 °C. Again, according to institutional preference, HIPEC regimens were mitomycin C, oxaliplatin, cisplatin, and others (5-fluorouracil, doxorubicin, etoposide, irinotecan) used alone or in combination. The volume of disease was recorded using the peritoneal cancer index described by Sugarbaker et al., which scored from 0 to 3 for each of the 13 divided regions of the abdominal and pelvis; thus, the scores range from 1 to 39 [54]. The PCI was scored during the cytoreductive surgery and extracted from the operative reports. When the PCI was not recorded prospectively, it was estimated from operation records and/or pathological reports. The SB-PCI, which has a minimum score of 0 and a maximum of 12, was recorded separately for each case. All sites and volumes of residual tumor following CRS were described using the Sugarbaker completeness of cytoreduction (CC) score: CC-0, no macroscopic residual tumor; CC-1, residual tumor deposits < 2.5 mm; CC-2, residual tumor deposits between 2.5 and 25 mm; and CC-3, residual tumor deposits > 25 mm [54]. Postoperative major complications were defined as any adverse event with a grade ≥ III according to the Clavien–Dindo classification system [55].

Patients were followed with clinical examinations and surveillance imaging according to institutional guidelines. This study was performed according to institutional ethical guidelines for medical research. Board approval was obtained for this retrospective study.

### 4.2. Statistical Analysis

Continuous measures were reported as mean (standard deviation) if they had a normal distribution, or as median (IQR) if they did not. The frequency and percentage of categorical data were calculated. The Kaplan–Meier method was used to calculate the OS and RFS. The OS was measured from the date of the patient’s first CRS for peritoneal metastases to death or final follow-up. The RFS was determined as the time interval from CRS to recurrence or last follow-up, which included death. If patients remained alive at the end of the follow-up period, data were censored.

## 5. Conclusions

This study revealed the clinical characteristics of long-term survivors undergoing CRS with or without HIPEC for peritoneal metastases from CRC. The long-term survivors tended to exhibit a low PCI/SB-PCI and to achieve a CC-0. In contrast, the fact remains that some of the long-term survivors revealed the factors considered to have adverse effects on survival outcome. Curative intent treatments such as CRS combined with perioperative chemotherapy, when feasible, should be performed even if patients have characteristics associated with poor prognosis. Further research is needed to confirm what prognostic factors have a significant influence on long-term survival and cure.

## Figures and Tables

**Figure 1 cancers-13-02964-f001:**
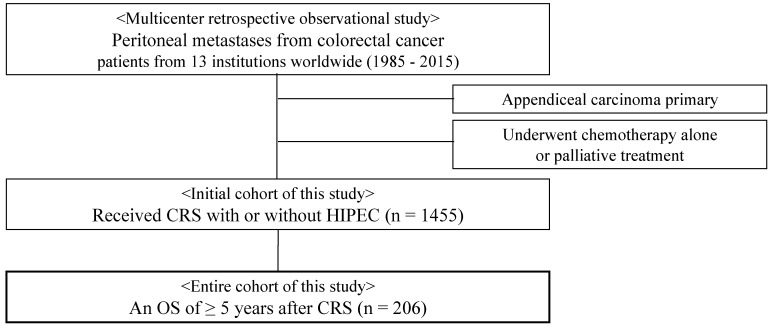
Flow diagram of patient enrollment. **Abbreviations:** CRS, cytoreductive surgery; HIPEC, hyperthermic intraperitoneal chemotherapy; OS, overall survival.

**Table 1 cancers-13-02964-t001:** Baseline characteristics.

Variable	Long-Term Survivors	Cured Patients
	(*n* = 206)	(*n* = 84)
Age, y, median (IQR)	58 (49–66)	55 (44–64)
Gender		
Male	101 (49.0%)	40 (47.6%)
Female	105 (51.0%)	44 (52.4%)
ASA grade		
I	97 (47.1%)	46 (54.8%)
II	84 (40.1%)	30 (35.7%)
III	6 (2.9%)	1 (1.2%)
Missing	19 (9.2%)	7 (8.3%)
Date of CRS		
Before 2001	16 (7.8%)	5 (6.0%)
Between 2001 and 2010	76 (36.9%)	37 (44.0%)
2011 or later	114 (55.3%)	42 (50.0%)
Onset		
Synchronous	96 (46.6%)	42 (50.0%)
Metachronous	96 (46.6%)	35 (41.7%)
Missing	14 (6.8%)	7 (8.3%)
Location of primary tumor		
Right colon	90 (43.7%)	38 (45.2%)
Left colon	101 (49.0%)	42 (50.0%)
Rectum	14 (6.8%)	4 (4.8%)
Missing	1 (0.5%)	0 (0%)
Histology		
Well to moderately	149 (72.3%)	67 (79.8%)
Mucinous	50 (24.3%)	15 (17.9%)
Poorly or signet ring cell	6 (2.9%)	2 (2.4%)
Missing	1 (0.5%)	0 (0%)
pT category		
pT ≤ 3	89 (43.2%)	33 (39.3%)
pT4	100 (48.5%)	45 (53.6%)
Missing	17 (8.3%)	6 (7.1%)
pN category		
N0	64 (31.1%)	25 (29.8%)
N1/2	123 (59.7%)	51 (60.7%)
Missing	19 (9.2%)	8 (9.5%)
Extraperitoneal metastases		
None	177 (85.9%)	78 (92.9%)
Liver metastases	27 (13.1%)	5 (6.0%)
Lung metastases	2 (1.0%	1 (1.2%)
PCI, median (IQR)	4 (2–7)	3 (2–5)
0–5	129 (62.6%)	66 (78.6%)
6–10	40 (19.4%)	14 (16.7%)
11–15	15 (7.3%)	2 (2.4%)
16–20	8 (3.9%)	1 (1.2%)
≥21	4 (1.9%)	0 (0%)
Missing	10 (4.9%)	1 (1.2%)
SB-PCI, median (IQR)	0 (0–2)	0 (0–1)
0	130 (63.1%)	60 (71.4%)
1–4	50 (24.3%)	15 (17.9%)
≥5	9 (4.4%)	2 (2.4%)
Missing	16 (7.8%)	7 (8.3%)

Abbreviations: CRS, cytoreductive surgery; HIPEC, hyperthermic intraperitoneal chemotherapy; IQR, interquartile range; PCI, peritoneal cancer index; SB-PCI, small bowel PCI.

**Table 2 cancers-13-02964-t002:** Treatment factors.

Variable	Long-Term Survivors	Cured Patients
	*(n* = 206)	(*n* = 84)
Preoperative chemotherapy		
Not performed	72 (35.0%)	31 (36.9%)
5-FU	127 (61.7%)	51 (60.7%)
Oxaliplatin	65 (31.6%)	30 (35.7%)
Irinotecan	58 (28.2%)	18 (21.4%)
Antiangiogenic	54 (26.2%)	25 (29.8%)
Anti-EGFR	12 (46.2%)	4 (4.8%)
Others	5 (2.4%)	5 (6.0%)
Chemotherapy cycles		
>6 cycles	29 (14.1%)	8 (9.5%)
≤6 cycles	73 (35.4%)	29 (34.5%)
Missing	32 (15.5%)	16 (19.0%)
Completeness of cytoreductive score		
CC-0	180 (87.4%)	77 (91.7%)
CC-1	22 (10.7%)	7 (8.4%)
CC-2	2 (1.0%)	0 (0%)
Missing	2 (1.0%)	0 (0%)
Number of organs resected, median (IQR)	2 (1–3)	2 (1–3)
Number of peritoneal sectors resected, median (IQR)	3 (1–6)	3 (1–4)
HIPEC		
Done	151 (73.3%)	62 (73.8%)
Not performed	55 (26.7%)	22 (26.2%)
HIPEC agent		
Mitomycin-based	85 (56.3%)	42 (67.7%)
Oxaliplatin-based	63 (41.7%)	19 (30.6%)
Others	3 (2.0%)	1 (1.6%)
Major complication (Clavien-Dindo III–IV)		
None	166 (80.6%)	72 (85.7%)
Intra-abdominal	31 (15.0%)	9 (10.7%)
Extra-abdominal	7 (3.4%)	2 (2.4%)
Missing	2 (1.0%)	1 (1.2%)
Length of stay, median (IQR)	19 (15–31)	18 (14–27)
Postoperative chemotherapy		
Systemic chemotherapy	148 (71.8%)	58 (69.1%)
Intraperitoneal chemotherapy	7 (3.4%)	2 (2.4%)
Systemic chemotherapy + Intraperitoneal chemotherapy	1 (0.5%)	1 (1.2%)
Not performed	48 (23.3%)	21 (25.0%)
Missing	2 (1.0%)	2 (2.4%)
Status		
Alive	134 (65.0%)	78 (92.9%)
Dead	72 (35.0%)	6 (7.1%)

Abbreviations. CRS, cytoreductive surgery; HIPEC, hyperthermic intraperitoneal chemotherapy; IQR, interquartile range; PCI, peritoneal cancer index.

**Table 3 cancers-13-02964-t003:** Site of recurrence and treatment for recurrence.

Variable	Total Number
	(*n* = 122)
Site of recurrence	
Isolated	
Peritoneum	43 (35.3%)
Liver	12 (9.8%)
Abdominal wall	11 (9.0%)
Lung	9 (7.4%)
Lymph nodes	5 (4.1%)
Bone	1 (0.8%)
Multiple	
Peritoneum + other site(s)	36 (29.5%)
Others	5 (4.1%)
Treatment for recurrence	
Reoperation ± chemotherapy	70 (57.4%)
Chemotherapy	21 (17.2%)
Palliative therapy	5 (4.1%)
Unknown	26 (21.3%)

## Data Availability

All data are available without restriction. Researchers can obtain data by contacting the corresponding author.

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
