# Peer review of "The Characteristics of 206 Long-Term Survivors with Peritoneal Metastases from Colorectal Cancer Treated with Curative Intent Surgery: A Multi-Center Cohort from PSOGI"

_cancers, 2021, doi:10.3390/cancers13122964_

Round 1
Reviewer 1 Report
I confirm the comments of the previous review
Author Response
Thank you for giving us the opportunity to revise our paper. The peer reviewers' constructive comments have significantly improved our manuscript.
We revised our original manuscript in response to reviewer suggestions.
Please review our revised manuscript.
I appreciate the opportunity to consider our work.

Reviewer 2 Report
Manuscript Review cancers 1244472
In this revised manuscript, the authors present data describing the characteristics of long-term survivors following CRS/HIPEC for colorectal cancer. The manuscript has been edited to avoid inappropriate comparisons and provides a better description of the characteristics of those with long-term survival.
The following minor points are suggested:
- Page 6, lines 44-46. The authors in the preceding portion of the paragraph nicely describe that the vast majority of patients with long term survival have low PCI and >90% have a CC-0 resection. The authors then state that “more noteworthy is that some patients with high PCI and/or CC 1-2 resection obtained long-term survival and cure. These patients should not be systematically excluded from the curative procedure”. This seems contradictory to the point of the study – 0 patients with a CC-2 resection were cured and only 1 patient with a PCI of 16-20 and 0 with a PCI of >20 were cured. Thus, I would favor a more tempered interpretation of the data. For example, in the following paragraph the authors comment that SB-PCI > 0 may be a relative contraindication to surgery.
Author Response
Thank you for your allowing us to revise our paper.
Our paper has been improved thanks to the helpful comments
According to your comment, we revised our paper.
Kindly check our revised manuscript.
The “Track Changes” function is used to mark up revisions in the manuscript.
I appreciate your consideration of our work.
We agree with the suggestion, so we changed the relevant sentences in the Discussion.
Changes: The sentences in the Discussion section now read as follows: ‘On the other hand, some of the long-term survivors presented with high PCI and/or CC-1/2 in this study. If the curative intent treatments are considered reasonable, patients with these adverse prognostic factors would not necessarily have to be excluded. (Page 6, line 44 to 47)’

This manuscript is a resubmission of an earlier submission. The following is a list of the peer review reports and author responses from that submission.
Round 1
Reviewer 1 Report
Dears authors,
this is a good review performed in important centers for HIPEC and CRS worldwide.
I think that your comments about the utility of this paper are appropriated. It's a retrospective report and it has not any high scientific evidence (I agree with the several limitations reported in the page 7); however it can be useful for the surgical community to better understand the real importance of a good patient selection. These data also can be useful in improving future studies, finalized to better analyze every single perspective you highlighted.
Reviewer 2 Report
Manuscript Number: cancers-1163161
Title: The characteristics of 206 long-term survivors with peritoneal metastases from colorectal cancer treated with curative intent surgery: A multi-center cohort from PSOGI
Author: Kamada et al
Summary: Kamada et al. submit a manuscript from the PSOGI collaborative
The manuscript outlines the characteristics of patients who underwent CRS/HIPEC for colorectal cancer with peritoneal metastasis. The group includes 13 institutions who perform CRS/HIPEC internationally. The study attempts to address an important question – what are the factors associated with long term survival following CRS/HIPEC for CRS/PM? The article is overall well-written. However, there are major methodological issues to be addressed prior to publication. Specifically, the following major points need to be addressed. Minor points are given for the authors’ consideration but are not required.
Major Points:
- This study does not have an appropriate comparison group which makes interpretation of the data nearly impossible. This is a retrospective study and understandably does not have a predefined control group. However, lacking the comparison to the group of patients who did NOT have long-term survival is a major methodological hurdle. For example, this data would suggest that patients with small volume disease (low PCI) and low SB-PCI scores would be more likely to have long term survival, but there is no comparison of groups. The authors attempt to make some comparison in the discussion between the HIPEC and no-HIPEC groups among long-term survivors. This is an interesting comparison and the manuscript could be reframed to address this question, but the results currently do not make this comparison.
The authors state in the discussion that ‘these data indicate that low PCI and CC-0 are associated with long-term survival and cure in patients with peritoneal metastases from CRC’. This is likely true based on other data, but how can the authors claim this from the current study? What is the comparison rate of low PCI and CC-0 resection in the group of non-long-term survivors? I find the assertion that lower SB-PCIs are needed for long-term survival to be similarly problematic – this is probably true, but there is not data in this manuscript to support the comparison that low SB-PCI have a higher cure rate than high SB-PCI patients.
The authors do note that this is a limitation of the study and indicate that they were not able to obtain data on all of the patients undergoing CRS/HIPEC for CRC/PM. Unfortunately, if this is true, the limitation precludes the assessment the authors are trying to make.
- The time frame for this study span 30 years. The highest volume center reported 83 cases during this time frame, which is less than 3 cases per year of the study. The majority of centers contributed less than one case per year to the cohort. From prior publications with this database, I suspect all these centers are high volume peritoneal disease institutions. However, from THIS data presented, there is no way to infer how many patients are seen at the institution.
- In Figure 1, the numbers of patients included are not listed. This needs to include the overall number of patients with CRC/PM in the database, the number of patients with CRS +/- HIPEC and then the final number of 206.
- Please include percentages in table 2.
Minor Points:
- Line 31 in the Results section is awkwardly worded. I think it should read “In 84 of the 206 patients, a cure was observed”; or something similar to that.
Reviewer 3 Report
This is a descriptive report of 206 patients who had CRS w/wo HIPEC and survived at least 5 years. Some of them did not have any recurrence and are thus considered "cured". The authors state that it was not possible to know or study the entire group of patients having had CRS at these hospitals to contrast with. This is a serious drawback and limits the generalizability of the results. All patients are selected in medicine, but the extent of selection of these 206 patients is not sufficiently known.
Literature on the treatment of peritoneal metastases and their treatment is extensive but mostly not conclusive although attempts to prospective trials have been done and reported. So we don´t lack evidence. This article adds limited new information. Most patients surviving that long have favourable characteristics, a not surprising finding. It is also not surprising that few had adverse signs. But how many of those with adverse signs do survive long and are "cured". Anecdotes are always there, but we can not base what we do on the anecdotes.
Otherwise, given these limitations, the article is well written and I have no specific comments that would improve writing more than knowing more about the non-included patients. At least at some or most of the 13 included hopsitals.